# Smoking Abstinence Self-Efficacy, Decisional Balance, and Quitting Desire Among Adult Smokers in Saudi Arabia: Gender-Based Cross-Sectional Study

**DOI:** 10.3390/healthcare13172158

**Published:** 2025-08-29

**Authors:** Samiha Hamdi Sayed, Olfat Abdulgafoor Gushgari, Fadiyah Abdullah Alshwail, Hanan Abd Elwahab Elsayed, Hanem Awad Mekhamier, Ebtesam Abbas Elsayed

**Affiliations:** 1Public Health Department, College of Health Sciences, Saudi Electronic University, Riyadh 11673, Saudi Arabia; o.gushgari@seu.edu.sa (O.A.G.); f.alshwail@seu.edu.sa (F.A.A.); e.elsayed@seu.edu.sa (E.A.E.); 2Community Health Nursing Department, Faculty of Nursing, Damanhour University, Damanhour 22516, Egypt; 3Assistance Medical Science Department, Applied College, University of Tabuk, Tabuk 71491, Saudi Arabia; helsayed@ut.edu.sa; 4Family and Community Health Nursing Department, Faculty of Nursing, Damietta University, Damietta 34511, Egypt; ham04@du.edu.eg; 5Community Health Nursing Department, Faculty of Nursing, Ain Shams University, Cairo 11517, Egypt

**Keywords:** smoking, quitting desire, abstinence self-efficacy, decisional balance, adults, Saudi Arabia

## Abstract

Background: Smoking is a major public health concern in Saudi Arabia, with significant gender differences influencing smoking behavior and cessation. Aim: This study aimed to investigate smoking abstinence self-efficacy (ASE), decisional balance (DB), quitting desire, and their predictors among adult male and female smokers in Saudi Arabia. Methods: A cross-sectional study was conducted using a convenience sample of 375 male and 220 female adult smokers recruited via social media. Data were collected through an online survey assessing personal health, smoking behavior, desire to quit, ASE, and DB. Logistic regression was used to identify predictors of earnest quitting desire, high ASE, and negative DB. Results: Males were more likely to smoke for 10 or more years (70.7% vs. 29.1%), maintain regular smoking patterns (86.9% vs. 54.1%), and exhibit high nicotine dependence (29.3% vs. 6.4%) compared to females. A higher proportion of females (76.8%) than males (66.9%) expressed a strong desire to quit. ASE was generally higher in males, with 49.6% showing average levels, while 46.4% of females had low ASE, particularly in social and positive mood contexts. Females displayed a higher prevalence of negative DB (73.6% vs. 58.1%), indicating greater awareness of smoking’s drawbacks. Both genders acknowledged the cons of smoking, though males perceived fewer pros. Conclusions: A complex interplay of factors influences smoking behavior and cessation among adult smokers. Gender differences also play a crucial role in smoking cessation factors among Saudi adults. Tailored cessation strategies addressing self-efficacy and motivation are recommended to enhance quitting success.

## 1. Introduction

Tobacco use is a major preventable cause of death and a significant public health issue. Smoking, the most widespread way of using tobacco, involves burning tobacco and inhaling or exhaling the smoke through products such as cigarettes, pipes, and cigars [1]. A specific smoking method is waterpipe use, where tobacco smoke is filtered through water before inhalation. Evidence shows that tobacco has no safe exposure level, and all forms of its use are harmful. Moreover, tobacco products containing nicotine are highly addictive [1,2].

According to the World Health Organization (WHO) estimates from 2020, tobacco use affects 22.3% of the global population and is responsible for killing about half of its users [2]. Tobacco smoking, both direct and indirect (secondhand), causes over eight million deaths annually, with nearly 1.3 million deaths attributable to secondhand smoke exposure [1]. Over the past 50 years (1970–2020), adult smoking prevalence has declined relative to overall population growth; among males, it remains high at around 32.6%, while among females, it is significantly lower at about 6.5%. However, this decline is uneven globally, with Asia and Africa showing the least decline in smoking prevalence [3].

In 2023, a Saudi national health survey found that 25% of men aged over 15 were active smokers, compared to much lower rates among women (3.3% Saudi women and 5% foreign resident women). Processed cigarettes were the most used tobacco product for both sexes (66% of men, 53.1% of women). Shisha and electronic cigarettes ranked second, more common in women (30.3% and 16.0%) than men (19.1% and 11.2%). Other tobacco forms, like cigars, rolled cigarettes, and pipes, were rare among men, each under 1% [4].

Smoking is a significant cause of chronic diseases, especially cancers, due to over 7000 harmful chemicals in tobacco smoke (including about 69 carcinogens) [5]. While lung cancer in smokers used to be mainly squamous cell carcinoma and small cell carcinoma, adenocarcinoma rates have risen, linked to RNA changes from smoking [6]. In addition, women who smoke often develop more severe Chronic Obstructive Pulmonary Disease (COPD) than men, potentially due to genetic factors, smaller airways, and higher oxidative stress [7]. Sex hormones (estrogen, testosterone, and androgens) also influence lung development, tobacco metabolism, and disease responses, contributing to gender differences in susceptibility, disease progression, and outcomes [6,7]. Smoking also increases the risk of stroke and atherosclerotic cardiovascular disease by triggering low-density lipoprotein oxidation, inflammation, and blood clots [8]. Male smokers face higher risks of heart disease, stronger nicotine dependence, and co-addictions [9] (e.g., alcohol, marijuana, illicit drugs) [10]. It also harms reproductive health—in men, reducing semen volume, sperm quality, and count [11] and in women, lowering ovarian reserve and disrupting menstrual cycles [12]. During pregnancy, smoking raises the risk of low birth weight, birth defects, perinatal death, and long-term health issues (e.g., respiratory disease, asthma, obesity, and bone fractures) [13].

Most smokers know the health risks, but while about half try to quit each year, fewer than 10% succeed [14]; in Saudi Arabia, 40% reported a quit attempt in the past year [15]. Quitting success depends on factors like decisional balance (weighing pros and cons), self-efficacy, motivation, and self-regulation. Low self-regulation reduces the likelihood of attempting to quit or coping with cravings, withdrawal, and negative emotions. A strong desire to quit, combined with high Abstinence Self-Efficacy—confidence in staying smoke-free—is a major predictor of successful cessation [16,17].

Self-efficacy, a key concept in behavior-changing models like the Health Belief Model [18], Theory of Planned Behavior [19], and transtheoretical model [20], significantly raises the likelihood of quitting smoking. It interacts with other factors such as perceived temptation and self-regulation strategies to support sustained behavior change [21,22]. Low self-efficacy is the primary cause of smoking relapse, acting as a mediator of relapse risk factors in models like the dynamic regulatory feedback [23]. During high-risk situations (e.g., exposure to smoking cues or negative emotions), drops in self-efficacy contribute to lapses. While the exact mechanisms remain underexplored, low self-efficacy undermines the smoker’s ability to manage challenging situations and recover from lapses [24,25]. Consequently, boosting self-efficacy is a central goal of cognitive–behavioral smoking cessation treatments [25,26].

Decisional balance (DB)—a key concept in the transtheoretical model—assesses readiness to quit smoking by weighing its perceived benefits (e.g., stress relief, social connection, weight control) against its drawbacks (e.g., health risks, financial costs, social disapproval) [20,27]. DB shapes attitudes toward smoking and influences motivation to quit [28]. Willingness to quit acts as a link between DB and quitting plans, with emotional support enhancing this effect [29]. Higher DB scores correlate with stronger motivation, greater self-efficacy, and a higher likelihood of starting cessation [22]. Conversely, lower DB scores indicate more barriers, weaker motivation, and less confidence to overcome challenges [30,31].

Understanding the factors driving smokers’ desire to quit is key to creating targeted, patient-centered cessation programs. Nurses, as frontline healthcare providers, play a crucial role by assessing smoking habits, educating patients, guiding quit attempts, collaborating with other professionals, and preventing relapse. Common nursing strategies include the 5A method and motivational interviewing, delivered through pamphlets, in-person counseling, and direct cessation advice [32].

Saudi Arabia’s healthcare system has made significant progress by offering various free smoking cessation services, such as the Quitline [33]. However, clinics report high relapse rates, underscoring the need for more comprehensive programs with regular follow-up [34,35]. In Jeddah, 57% of smokers want to quit, highlighting the importance of understanding the factors that influence cessation to design tailored interventions, strengthen support, and reduce relapses [36]. Thus, this study examines quitting desire, ASE, and DB among adult male and female smokers in Saudi Arabia, exploring factors that influence each. It aims to deepen the understanding of smoking cessation decisions, aiding healthcare professionals in designing client-centered cessation services to support smokers better and reduce relapse rates. Notably, the study highlights gender differences in quitting desire, ASE, and DB, providing a valuable knowledge foundation for future research and more detailed analyses.

## 2. Materials and Methods

### 2.1. Design and Setting

This study employed a social media-based cross-sectional design, adhering to the Strengthening the Reporting of Observational Studies in Epidemiology (STROBE) guidelines.

### 2.2. Study Participants and Sampling

The target group was adult males and females in Saudi Arabia (≥20 years) who had well-established smoking habits. They were recruited using specific eligibility criteria: being smokers for more than six months (regular/daily or irregular/nondaily smokers), using any smoking type (cigarettes, e-cigarettes, hookah, e-hookah), and being keen to participate in the study.

The convenience sampling technique was adopted to select the participants to fulfill the required sample size that was determined using the following equation parameters: 95% confidence level (Z_α/2_ = 1.96 for alpha 0.05), proportion of adult male and female smokers in Saudi Arabia (*p* = 26.3%), based on the WHO 2020 estimates [37], design effect of sampling (*D* = 2), and margin of error (*E* = 0.05). These parameters resulted in a sample size of 595 male and female smokers.$$\frac{{Z^{2}}_{\alpha /2}\times P\times \left(P-1\right)\times 2}{E^{2}}$$The researchers used SurveyMonkey software, version 3.3.8 (SurveyMonkey Inc., San Mateo, CA, USA) to define the participants’ search criteria (all regions of Saudi Arabia, adult age, and both males and females) and social media platforms (such as Twitter, LinkedIn, and Instagram) to select a representative sample. Participants were recruited through posts shared on institutional and organizational social media pages associated with the research team, as well as through personal accounts of collaborators to broaden the reach. No paid advertisements or direct message solicitations were used during recruitment. Additionally, the eligibility criteria are applied at the start of the survey through screening questions about age, smoking status, and duration (the survey is declined if irrelevant). Numerous survey rounds were conducted to guarantee that it was disseminated throughout multiple networks to maximize diversity in participants’ age, background, and regions, thereby decreasing bias and increasing representativeness.

### 2.3. Survey Development

The researchers designed a structured survey using credible literature. It contained four sections:

Basic Personal Data and Health History:

Age, gender, marital status, nationality, educational level, residence, perceived income adequacy, working status, and self-reported psychological or physical health problems.

Smoking Behavior-Related Data:

Smoking initiation time, duration, preferred type, perceived dependency, causes of smoking initiation, presence of a smoking member in the family, number of previous quitting trials, and prior use of nicotine replacement therapy. A yes or no question was also asked about the earnest desire to quit [23,24].

Smoking Abstinence Self-Efficacy (ASE) Scale:

It is a valid tool developed by Velicer et al. (1990) to assess the individual’s ability to resist smoking desire on three different stimulating occasions [38]. It has nine items distributed over three domains, reflecting the individual capacity to withstand the impact of negative affect (3 items, numbers 3, 6, and 9), social/positive mood (3 items, numbers 1, 4, and 7), and habitual craving (3 items, numbers 2, 5, and 8) on smoking desire [38]. It had satisfactory internal consistency reliability: negative affect (α = 0.95), social/positive (α = 0.93), and habitual craving (α = 0.92) [38]. The weighted mean scores were used to analyze the smoking ASE items: not confident (1.00–1.80), slightly (1.81–2.60), moderately (2.61–3.40), very (3.41–4.20), and extremely (4.21–5.00) confident. Total and subscale scores were further categorized into three levels: low (≤3.39), moderate (3.40–3.79), and high (≥3.80) [39].

Decisional Balance (DB) Scale:

It had six items assessing the individual’s perceived smoking pros/advantages (3 items numbers 1, 3, and 5) and cons/disadvantages (3 items—numbers 2, 4, and 6). It was first developed by Velicer et al., and then its internal consistency was validated by Ward et al. [40,41]. It demonstrated strong factorial invariance across demographic variables, with a satisfactory coefficient alpha ranging between 0.51 and 0.67 across all groups [40]. The participants rated the importance of every item to their smoking decision on a 5-point Likert scale, ranging from “unimportant” (1) to “extremely important” (5). The weighted mean scores were used to analyze the DB scale items: unimportant (1.00–1.80), slightly (1.81–2.60), moderately (2.61–3.40), very (3.41–4.20), and extremely (4.21–5.00) important. The total pros and cons were categorized as low (≤3.39), average (3.40–3.79), and high (≥3.80) [39]. Moreover, the total DB score was calculated by subtracting the total scores of cons from the pros and classifying them as negative or positive.

### 2.4. Survey Validity and Reliability

The researchers translated the study scales into Arabic using the DeepL Translator software, version 24.11.4.14424 (DeepL SE Co., Cologne, NW, Germany). An expert researcher performed a back translation to confirm its precision. A panel of five experts reviewed and approved the survey content by examining the wording, ranking, and scoring of the items. The researchers modified the survey based on panel feedback, demonstrating an appropriate Content Validity Index (CVI = 0.862). The survey’s reliability was guaranteed by Cronbach’s Alpha coefficient test (α), with satisfactory results (ASE = 0.82, DB = 0.81). The survey was piloted on 10% of the sample size (omitted from the study sample) to ensure its precision, wording, and pertinence. Consequently, the appropriate modifications were completed.

### 2.5. Data Collection

After obtaining ethical approval, the SurveyMonkey program (SurveyMonkey Co., San Mateo, CA, USA) was utilized for data collection. The researchers circulated the digital survey link using specific social media platforms (Twitter, LinkedIn, Instagram) because of their broad and diverse user bases in Saudi Arabia, targeting different adult demographics.

To validate user identities, several measures were implemented: The survey itself gathered demographic and behavioral data consistent with actual participants, and the invitation contained explicit instructions and eligibility requirements (regarding age, smoking status, and duration). Although the authors did not incorporate direct automated detection algorithms for social media accounts, data quality checks were conducted to find duplicate or inconsistent responses, which were then removed from the analysis. To avoid automated or bot accounts from Twitter, the researchers used follower selection and targeted advertising to help limit exposure to non-human users; the combination of survey controls helped lower the possibility of algorithmic or bot participation.

The data was collected over three months, from October 1 to December 30, 2024. The survey’s average completion time was 7–10 min, with a 91.0% response rate. The SurveyMonkey platform’s built-in features help enhance data quality. It automatically tracks IP addresses and timestamps to help identify potential duplicate responses by screening and excluding duplicates. Additionally, all the required fields for essential questions were set to minimize incomplete responses. Entries with less than 80% completion or missing key data were excluded from analysis to ensure data integrity.

### 2.6. Statistical Analysis

The researchers used IBM Statistical software, version 27 (IBM Corp., Armonk, NY, USA). The researchers used descriptive statistics to summarize numerical and categorical variables. The Shapiro–Wilk test (*p* > 0.05) was used to assess the normality of the data. The statistical significance of gender differences for categorical variables was judged using Chi-square or Fisher’s exact tests. The weighted mean scores were used to analyze the smoking ASE and DB scale items. The Analysis of Variance (ANOVA) test was used to analyze the mean differences in the total scores of the ASE and DB scales based on the participants’ personal and smoking-related data. The Chi-square test was used to test the significant differences between males’ and females’ earnest desire to quit based on their personal and smoking-related data.

Logistic regression analysis was employed to investigate the predictors of earnest quitting desire, high smoking ASE, and negative DB. Univariate logistic regression was first performed to estimate the odds ratios (ORs) and 95% confidence intervals (95% CIs) for the studied variables. The OR was then adjusted using multivariate logistic regression analysis while controlling for the confounding effects of nationality, marital status, and working status. The coefficient of determination, as represented by the Cox and Snell R^2^ and Nagelkerke R^2^ values, was used to estimate the model’s fitness. The Chi-square test was used to judge the model’s significance (*p* < 0.05). The model’s goodness of fit was confirmed via the Hosmer and Lemeshow Test; a non-significant *p*-value (*p* > 0.5) signifies model fitting for data [42]. The cut-off point of statistical significance (*p*-value) was < 0.05.

### 2.7. Ethical Considerations

The researchers adhered to the Declaration of Helsinki to ensure respect for individuals, their right to informed consent, and the prioritization of participant welfare over scientific or societal interests. The researchers obtained Institutional Review Board approval from the Saudi Electronic University’s research ethics committee on 18 June 2023 (SEUREC-4457). The respondents were provided with an adequate explanation of the study’s aim and importance, and informed digital consent was initially obtained from each respondent. Consent was implied by the participants’ choice to continue with the survey. The researchers assured the respondents that their data would be kept anonymous, confidential, and used only to serve the scientific research objectives. The respondent’s right to voluntary participation and withdrawal without repercussions was also clarified.

## 3. Results

### 3.1. Personal Characteristics of Study Population

Table 1 shows that the highest percentages of males (57.3%) and females (75.4%) are aged between 20 and 30 years, with a mean age of 27.16 ± 0.501, compared to 30.52 ± 0.344 among males. Married status and bachelor’s education were prevalent among males (62.6%, 71.2%) and females (70.0%, 65.0%), respectively. Working males represent 77.1%, while 58.6% of females are students. Most males (93.3%) and females (94.5%) are Saudi nationals. Notably, 31.4% of females reside in the eastern region, compared to 23.5% of males. The highest percentages of males (47.2%) and females (53.6%) have an adequate monthly income. A low percentage of males (21.6%) and females (26.4%) has physical health problems (bronchial asthma, diabetes mellitus, heart diseases, obesity, stress colon, and iron deficiency anemia), which were frequently reported. However, 54.9% of males have psychological health problems compared to 33.2% of females (worry, stress, tension, and depressed mood were the most frequently reported). Significant differences were observed between males and females for all personal and health data (*p* < 0.05) except marital status, nationality, and physical health problems (*p* > 0.05).

### 3.2. Smoking Behavior of Study Population

Table 2 illustrates that most males (70.7%, 57.3%) have been cigarette smokers for 10 years or more, compared to 29.1% and 23.2% of females, respectively. The highest percentage of males have regular smoking patterns (86.9%) and extremely high smoking dependency (29.3%) compared to 54.1% and 6.4% of females, respectively. Stress is the highest reported cause of smoking initiation among males (41.1%) and females (39.5%). Most males (69.9%) and females (83.6%) have a smoking family member. The highest percentages of males (41.3%) and females (37.3%) had 1 to 5 previous quitting attempts, and 81.1% of males and 96.8% of females had not previously used nicotine replacement therapy. Most males (79.5%) and females (73.6%) earnestly desire to quit. There are significant differences between males and females for all smoking behavior-related data (*p* < 0.05) except the number of previous quitting trials (*p* > 0.05).

### 3.3. Item and Total Scores of Smoking ASE and DB of Study Population

Table 3 shows that the overall weighted mean score of the smoking ASE scale among males was 3.22 ± 0.045, compared to 3.02 ± 0.075 for females. The weighted mean score of the negative affect subscale (items 3, 6, 9) in males is 3.34 ± 0.058 compared to 2.97 ± 0.087 in females. The weighted mean score of the social or positive mood subscale (items 1, 4, 7) for males is 3.33 ± 0.045, compared to 2.85 ± 0.082 for females. The weighted mean score of the habitual craving subscale (items 2, 5, 8) is 2.96 ± 0.056 for males compared to 3.27 ± 0.086 for females. There are significant differences between males and females for the total smoking ASE scale and its three subscales (*p* < 0.001).

Table 4 illustrates that the overall weighted mean score of the smoking DB scale in males is 3.23 ± 0.033, compared to 2.63 ± 0.066 in females. The weighted mean score of perceived pros (items 1, 3, and 5) for males is 3.27 ± 0.059, compared to 2.53 ± 0.086 for females. The weighted mean score of perceived cons (items 2, 4, and 6) for males is 3.18 ± 0.056, compared to 2.71 ±0.084 for females. There are significant differences between males and females for the total smoking DB scale and its pros and cons subscales (*p* < 0.001).

Table 5 depicts that 49.6% of males have an average ASE, while 46.4% of females have a low level. More than two-thirds (64.5%) of males have average ASE during negative affect situations, while 53.6% of females have low levels. More than two-thirds (65.5%) of females have low ASE during social or positive mood situations compared to 52.8% of males. Moreover, 68.3% of males have low ASE during habitual craving situations compared to 43.6% of females. There are significant differences between males and females for the total ASE and its three subscales (*p* < 0.001).

Regarding the smoking DB, the highest percentages of females (73.6%) and males (58.1%) have negative DB. Additionally, 72.8% of males and 53.6% of females perceived low pros of smoking; however, 63.7% of males and 66.4% of females perceived high cons of smoking. There are significant differences between males and females for the total DB scale and pros and cons subscales (*p* < 0.001).

### 3.4. Factors Associated with Smoking ASE, DB, and Earnest Quitting Desire

Table 6 shows significant mean differences in the total ASE mean scores based on all personal and smoking-related data between males and females (*p* < 0.05) except for marital status, nationality, and presence of physical health problems (*p* > 0.05)

Based on all personal and smoking-related data, males and females have significantly different overall DB mean scores (*p* < 0.05), except for marital status, nationality, monthly income, physical health issues, family members who smoke, quitting trials, and use of nicotine replacement therapy (*p* > 0.05) (Table 6).

Table 7 reveals statistically significant differences in the earnest desire to quit smoking between males and females (*p* < 0.05) based on all personal and smoking-related data, except for marital status, monthly income adequacy, and the presence of physical health problems (*p* > 0.05).

### 3.5. Predictors of Negative DB, High Smoking ASE, and Earnest Quitting Desire

Table 8 portrays that young age (31–40) (AOR = 0.20, 95%CI = 0.07–0.59, *p* = 0.004), male gender (AOR = 0.05, 95%CI = 0.02–0.11, *p* = 0.000), and living in the central (AOR = 0.46, 95%CI = 0.09–0.88, *p* = 0.024) and eastern (AOR = 0.21, 95%CI = 0.13–0.81, *p* = 0.000) regions are negative predictors of negative DB. However, bachelor education (AOR = 1.54, 95%CI = 0.14–3.11, *p* = 0.003), long smoking duration (5–9 years) (AOR = 1.31, 95%CI = 1.07–3.24, *p* = 0.006), regular smoking (AOR = 2.79, 95%CI = 1.53–3.24, *p* = 0.000), high smoking dependency (AOR = 2.67, 95%CI = 1.41–3.69, *p* = 0.005), using nicotine replacement therapy (AOR = 2.19, 95%CI = 1.34–3.58, *p* = 0.002), low ASE (AOR = 8.33, 95%CI = 3.89–15.17, *p* = 0.000), average ASE (AOR = 4.68, 95%CI = 2.19–10.03, *p* = 0.000), and earnest quitting desire (AOR = 2.75, 95%CI = 1.35–3.61, *p* = 0.022) are positive predictors of negative DB.

Concerning the ASE, living in the eastern region (AOR = 0.29, 95%CI = 0.11–0.92, *p* = 0.000), having psychological health problems (AOR = 0.74, 95%CI = 0.11–0.97, *p* = 0.014), smoking a hookah (AOR = 0.12, 95%CI = 0.02–0.85, *p* = 0.034), perceived high smoking dependency (AOR = 0.58, 95%CI = 0.13–0.74, *p* = 0.021), number of previous quitting trials in the last year (6–10 trials) (AOR = 0.27, 95%CI = 0.08–0.91, *p* = 0.035), and negative DB (AOR = 0.10, 95%CI = 0.04–0.45, *p* = 0.035) were negative predictors of high ASE. Conversely, bachelor’s education (AOR = 2.55, 95%CI = 1.26–4.11, *p* = 0.021), short smoking duration (<one year) (AOR = 1.41, 95%CI = 1.06–0.76, *p* = 0.008), and earnest quitting desire (AOR = 2.44, 95%CI = 1.03–5.81, *p* = 0.044) are positive predictors of ASE (Table 8).

Regarding earnest quitting desire, smoking hookah (AOR = 0.06, 95%CI = 0.01–0.58, *p* = 0.030), regular smoking (AOR = 0.17, 95%CI = 0.05–0.55, *p* = 0.003), perceived high smoking dependency (AOR = 0.11, 95%CI = 0.04–0.30, *p* = 0.003), using nicotine replacement therapy (AOR = 0.08, 95%CI = 0.02–0.34, *p* = 0.000), low ASE (AOR= 0.60, 95%CI = 0.24–0.72, *p* = 0.021), and negative DB (AOR = 0.21, 95%CI = 0.08–0.58, *p* = 0.021) were negative predictors of earnest quitting desire. However, having physical health problems (AOR = 1.48, 95%CI = 1.06–2.91, *p* = 0.023) and previous quitting trials in the last year (6–10 trials) (AOR = 1.23, 95%CI = 1.08–4.07, *p* = 0.023) were positive predictors of serious quitting desire (Table 8).

## 4. Discussion

The current study shows that male participants exhibited more significantly entrenched cigarette smoking habits (70.7% vs. 29.1% for females), while hookah smoking is more common in females (38.6% vs. 15.5% for males). Males also have a significantly higher smoking duration of 10 years or more (70.7% vs. 29.1% for females) and reported higher smoking dependency (29.3% vs. 6.4% for females). Most males and females have smoking family members, have no physical health problems, report 1 to 5 previous quitting trials, and have not previously use nicotine replacement therapy. However, significantly higher psychological health issues were reported among males (54.9%) than among females (33.2%). Stress and worry were the most reported psychological problems that may also be exacerbated by smoking, creating a vicious cycle that makes quitting more challenging. Interventions aimed at improving mental health could, therefore, play a central role in smoking cessation programs, especially for males.

These findings are consistent with global trends showing higher smoking prevalence and intensity among males [36]. Many studies have also demonstrated congruent national trends. Alnasser et al. showed that males smoked significantly more cigarettes (56.2%) than females (23.5%), while females smoked more hookah/shisha (67.6%) than males (24.7%) [43]. Abdelwahab et al. also reported similar patterns of smoking duration and intensity among males in Saudi Arabia [44]. Qattan et al. corroborate these findings and elaborate that gender is significantly associated with smoking intensity, suggesting that gender-specific factors continue to influence smoking patterns in the region. Stress was the primary reported cause of smoking initiation for both genders (41.1% males and 39.5% females) [45]. These findings align with global trends observed by Stubbs et al., who identified stress as a universal trigger for smoking initiation and continuation across various countries and cultures [46]. Moreover, Le et al. found that huge factors influencing smoking decisions, such as tobacco experimentation, risk perceptions, awareness, gender, exposure to media influence, social influence, financial status, and mental and physical health status, suggest regional variations in smoking onset motivations [47].

### 4.1. Earnest Desire to Quit

The current study described a significantly higher percentage of males (79.5%) than females (73.6%) who reported an earnest desire to quit. This high level of motivation is encouraging and suggests that there is a strong foundation for implementing effective smoking cessation programs in Saudi Arabia. The study also found that males significantly reported higher (18.9%) previous use of nicotine replacement therapy than women (3.2%) during previous quitting attempts. However, the number of previous quitting attempts was not significantly different between genders, indicating that both males and females face similar challenges in maintaining long-term abstinence. The gender difference in quitting motivation aligns with the global trends observed by Guimaraes-Pereira et al. [48]. Similarly, a Saudi study by Monshi et al. showed that most smokers were interested in quitting either cigarettes (58%) or waterpipes (17.1%) [49]. However, this contrasts with a Korean study by Hwang and Park, where gender differences in quitting desire were less pronounced [50]. This contradiction could be explained by poor smoking quitting intention among participants in this contradictory study, and most of them were males.

The present study shows that previous quitting attempts and physical health problems positively predicted earnest quitting desire. Similarly, Hwang and Park found that individuals who had earlier attempts to quit smoking were more likely to have a desire to quit smoking [50]. A Saudi study by Al-Nimr et al. found that the principal reason for readiness to quit was health concerns, while fear of mood changes was the most frequent reason for being reluctant to quit smoking [51]. Similarly, the present study supports this by showing that the absence of psychological health problems is significantly associated with the earnest desire to quit, with statistical significance between males and females.

The present study found that smoking hookah, regular smoking, high perceived smoking dependency, low ASE, and negative DB were negative predictors of earnest quitting desire. These findings align with a study by Khan et al., which found that shisha was higher than cigarette smoking among the participating Jeddah population, and most of them agreed that smoking is addictive in both forms. However, a minority believed that shisha was more addictive [36]. These beliefs might be a factor in Saudis’ adoption, continued use, and lack of desire to stop using hookah. According to Albasheer et al.’s study, conducted in Jazan, 71% of participants cited self-efficacy as their primary motivation for quitting smoking [52]. A study by Amer added that the likelihood of quitting smoking was significantly higher among Saudi smokers with high self-efficacy [53]. Lin et al. clarified that quitting behavior was significantly correlated with quitting intention among Chinese smokers in the low nicotine dependency group. Additionally, they emphasized the significance of taking smokers’ nicotine dependency level into account when devising smoking cessation plans [54]. Moreover, Cao et al. explained that smoking DB is associated with plans to stop smoking, and that this relationship is mediated by willingness to stop smoking [29].

### 4.2. Smoking Abstinence Self-Efficacy

The present study shows that males generally have slightly higher overall ASE than females, with 49.6% of males having average smoking ASE and 46.4% of females having low ASE. These findings suggest that males feel more confident in their capacity to withstand smoking compared to females, which aligns with previous research indicating that males often report higher self-efficacy in smoking cessation contexts [24]. A recent qualitative study by Lakshmi et al. emphasized self-efficacy or behavioral control as one of the main psychological capabilities to resist tobacco use initiation [55].

The ASE subscale analysis reveals that males demonstrate significantly higher ASE during negative affect situations (64.5% average ASE) than females (53.6% low ASE). This finding is particularly noteworthy and may reflect cultural norms regarding emotional expression and coping mechanisms. It also reflects that males seem to have more self-confidence to withstand smoking when experiencing negative emotions, while females struggle more in these scenarios. A qualitative study in the Netherlands by Dieleman et al. reported that psychological factors like emotion and stress were the primary obstacles to smoking cessation in females. In contrast, environmental factors are the main ones in men [56]. The present study also showed that both genders have low ASE in social or positive mood situations (e.g., being at a party or relaxing with friends), with significant differences between males (52.8%) and females (65.5%). This finding suggests that both genders struggle to resist smoking in social settings, but females face even more significant challenges. Social smoking is often reinforced by cultural norms and peer pressure, especially since most males and females have a smoking family member. These findings align with the findings of Alqahtani et al. [57] and Lakshmi et al. [55], who emphasized the role of social context in sustaining and reinforcing smoking behaviors.

The present study finds that a significantly higher percentage of males (68.3%) has low ASE during habitual craving situations than females (43.6%). This finding underlines the difficulty of overcoming habitual smoking behaviors, which are often deeply ingrained and triggered by environmental cues. It also highlights the challenge of overcoming nicotine addiction, a finding similar to global trends reported in the WHO report on tobacco use [37]. Likewise, Blonde & Falomir-Pichastor found that highly dependent smokers reported more smoking cravings and lower desire to quit than the less dependent group [58]. Behavioral interventions, such as cue exposure therapy or mindfulness-based approaches, could be effective in helping individuals manage cravings and break habitual patterns [59].

The current study provides valuable insights into the factors predicting ASE in Saudi Arabia. It shows that living in the eastern region, having psychological health problems, smoking alternative tobacco products such as shisha (hookah), perceived high smoking dependency, having multiple failed smoking quitting trials in the last year (6–10 trials), and having negative DB toward smoking were negative predictors of ASE, suggesting less likelihood of having self-confidence in their capability to abstain from smoking. These findings can inform targeted interventions and support strategies to quit smoking, which many recent studies have explained. In a Vietnamese study by Luu et al., the likelihood of successfully quitting may be lowered if one lives in an environment where smoking is accepted as the norm, is frequently exposed to pro-tobacco advertising, and is socially isolated [60]. Low self-control, negative moods, stress related to work or life, and tobacco dependence were all identified as negative factors for smoking cessation failure in a Chinese study by Wang et al. [61]. A comparative study in Britain by Richardson et al. explained that having a mental health problem was associated with the desire to quit, a heavy smoking rate, and a perceived struggle in enduring abstinence [62]. Moreover, a study by Rajan et al. showed that age and gender were not significantly associated with self-efficacy, which aligns with the current research. However, it showed that nicotine dependence is not significantly associated with self-efficacy, which can be attributed to a large proportion of their participants having low to moderate nicotine dependence [63].

The current study proved that bachelor’s education, shorter smoking duration, and earnest quitting desire are positive predictors of ASE among smokers, suggesting a greater likelihood of having confidence in their ability to abstain from smoking. Many studies supported these findings. Rajan et al. showed that education equips individuals with background knowledge and raises awareness about smoking-associated health risks [63]. A Chinese study by He et al. demonstrated that smokers with prolonged tobacco use had more difficulty in quitting due to higher dependency, which could explain the reason behind the increasing self-efficacy with short smoking duration found in this study [64]. Moreover, Alanazi et al. showed that ASE was associated with a greater desire to quit, and smoking negative consequences and reinforcement were associated with lower ASE. Thus, participants who had negative thoughts about smoking required greater self-efficacy to resist smoking [65].

### 4.3. Decisional Balance

The current study indicates that both genders predominantly have negative DB, with significantly higher differences among females (73.6%) than males (58.1%), indicating that they perceive more cons than pros in smoking. It is an encouraging finding for public health efforts and reflects an increased awareness of smoking risks. A person’s attitude toward smoking can be influenced by DB, which can predict behavior change [28]. Research indicates that those with low DB scores see more obstacles and are less motivated to change, which is frequently reflected in a lower level of self-efficacy—the belief that one can overcome these obstacles [30,31]. Evidence shows that increasing the DB score increases the quitting probability [22]. Moreover, Cao et al. found that the association between smoking DB and planning to quit smoking was mediated by willingness to quit, which was further moderated by emotional support [29].

Remarkably, a high percentage of males (72.8%) perceived low pros of smoking compared to 53.6% of females, with a high perception of smoking cons among both genders (63.7% for males, 66.4% for females), which is promising for cessation efforts. However, the persistence of smoking despite this awareness, coupled with the higher percentage of a negative DB, especially among females, suggests a more complex decision-making process among female smokers. The cognitive dissonance theory in smokers may offer insights into this phenomenon by providing a framework for recognizing the discrepancy between being knowledgeable about smoking harm and continuing to smoke. Thus, smokers are more likely to modify their beliefs to defend their actions than to stop smoking. Smokers are encouraged to justify their actions by endorsing additional positive views concerning smoking, and such attitudes systematically shift as smoking status changes [66]. Moreover, Ruffin et al. proved that nicotine’s addictive properties make it difficult for people to stop smoking, leading to chronicity of smoking and its associated health problems [67]. These findings underscore the need for gender-specific approaches in smoking cessation programs. Future interventions should leverage the negative DB and support the high smoking cons to motivate quit attempts.

Research indicates that females have lower success rates in quitting smoking, which may be partly due to NRT being less effective for them. This difference arises because males’ smoking behavior is primarily driven by the pleasurable effects of nicotine, resulting in a stronger response to NRT, while social and economic factors influence females’ smoking more. Although many technology-based cessation programs apply gender-neutral approaches, combining pharmacological treatments with strategies that address these socio-contextual challenges can better support women in quitting [68]. Future interventions tailored by gender, using technology, and targeting women’s unique barriers may improve quitting success for females [69].

The current study provides valuable insights into the predictors of negative DB. It found that young age (31–40), male gender, previous use of nicotine replacement therapy, and living in central or eastern regions were negative predictors of negative DB, suggesting that these groups may be less likely to perceive the cons of smoking. Bachelor’s education, longer smoking duration (5–9 years), regular smoking, high smoking dependency, and low and average ASE were positive predictors of negative DB, indicating that these factors may increase awareness of the negative aspects of smoking. Similarly, Sayed et al. portrayed that long-term regular smokers tend to be in the early smoking cessation stages (pre-contemplation), where the pros of smoking outweigh the cons, and they may have lower self-efficacy and DB for quitting. They also showed a positive moderate relationship between educational level and smoking stages, and that a higher self-efficacy for quitting is associated with a more favorable DB toward cessation [22].

Moreover, Cao et al. reported that a higher nicotine dependence level is related to a more favorable DB toward continued smoking among male smokers [29]. A recent study in Northern Ireland by Tate et al. reported lower odds of smoking susceptibility among individuals with fewer family smokers, access to information about smoking, higher levels of openness, self-reported well-being, self-efficacy and perceived behavioral control to quit, negative attitudes toward smoking, and fewer smoking friends [70]. Narimani et al. also showed that higher self-efficacy is associated with a more favorable DB toward quitting [34]. Gokbayrak et al. found that older smokers tend to have a more favorable DB toward quitting, where pros and cons are most important in preparing individuals to act or start quitting, but not in maintaining it. They added that emotional and social factors influence DB rather than just physical addiction [71]. Having helping relationships, workplace smoking bans, and other tobacco control policies can affect the smoking DB.

In conclusion, this study contributes significantly to understanding smoking behaviors, desire to quit, ASE, and DB among Saudi adult smokers, highlighting essential gender differences. When compared with recent studies, it becomes clear that these findings are part of broader global trends in smoking behaviors and cessation efforts. Future research should continue to explore these gender disparities and their implications for public health strategies in the region, with a focus on developing targeted, culturally sensitive interventions to address the specific needs of male and female smokers in Saudi Arabia. While many predictors align with international findings, some factors, particularly those related to regional differences and hookah use, appear more specific to the Saudi context. These insights can inform culturally tailored smoking cessation interventions in Saudi Arabia and contribute to the global understanding of smoking cessation behaviors.

### 4.4. Study Implications

The findings of this study have several implications for population health and smoking cessation programs and services in Saudi Arabia. First, gender-specific interventions are needed to address the unique challenges faced by male and female smokers. For males, interventions should focus on reducing psychological health problems and improving stress management skills. For females, interventions should address social and cultural barriers to quitting, such as stigma and lack of support. Second, the study highlights the importance of addressing habitual cravings, which were identified as a significant barrier to quitting for both genders. Behavioral interventions, such as the cognitive–behavioral approach and stress reduction using a mindfulness strategy, could be effective in helping individuals manage cravings and develop healthier coping mechanisms [59]. Finally, the high level of earnest quitting desire among both genders suggests a strong potential for successful smoking cessation programs in Saudi Arabia. However, these programs must be tailored to address the specific needs and challenges of different demographic groups, including those with lower education levels, psychological health problems, and high smoking dependency.

### 4.5. Limitations and Future Research

While this study provides valuable insights into gender-based differences in smoking behavior and cessation, it is not without limitations. The cross-sectional design limits the ability to establish causal relationships between variables. Future research should consider longitudinal studies to examine how ASE and DB change over time and their impact on successful quitting and better understanding the long-term effects of smoking cessation interventions. Additionally, qualitative research could provide deeper insights into the cultural and social factors influencing smoking behaviors and cessation attempts in Saudi Arabia, particularly among women, given the relatively recent increase in female smoking rates. The study also relied on self-reported data, which may be subject to bias and social desirability. Future research could incorporate objective measures, such as biochemical confirmation of smoking status, to improve the accuracy of the findings. Moreover, using a social media-based recruitment method may introduce selection bias, as it likely favors younger and digitally literate individuals, which may limit the generalizability of the findings to the broader population of adult smokers in Saudi Arabia. Future studies considering more diverse or randomized recruitment strategies to improve external validity are recommended.

## 5. Conclusions

The study revealed that males and females differ significantly in their smoking patterns, dependency levels, and reasons for smoking initiation. Males were more likely to be long-term smokers with higher dependency levels, while females tended to have shorter smoking durations and lower dependency. The study revealed that males had greater overall ASE scores than females, particularly in situations involving negative affect and social or positive mood. These findings suggest that males may feel more self-confident in their capacity to resist smoking in emotionally charged or social situations. However, both genders reported low ASE in habitual craving situations, indicating that cravings remain a significant barrier to quitting for both males and females. The study also revealed significant gender differences in smoking decisional balance. Despite these differences, most males and females earnestly desired to quit.

This study highlights the complex interplay of factors influencing smoking behavior and cessation in Saudi Arabia. By considering gender differences, cultural context, and regional variations, healthcare providers and policymakers can develop more effective, targeted interventions to reduce smoking prevalence in the kingdom. However, it is essential to note that the recruitment strategy relying on social media platforms may have introduced selection bias by favoring younger and more digitally literate individuals, which could limit the generalizability of these findings to the broader population of adult smokers in Saudi Arabia. Future research and interventions should thus aim to address these limitations by employing more diverse recruitment methods. Future research and interventions should focus on addressing the specific challenges faced by male and female smokers, enhancing ASE, and shifting the decisional balance further towards cessation.

## Figures and Tables

**Table 1 healthcare-13-02158-t001:** Male and female participants’ personal and health data.

Parameters	Categories	Males n (375)	Females n (220)	Test (*p*-Value)
		No. (%)	No. (%)	
Age (years)	- 20–30	215 (57.3)	166 (75.4)	χ^2^ = 21.182 (<0.001 **)
	- 31–40	126 (33.6)	40 (18.2)	
	- 41–50	34 (9.1)	14 (6.4)	
	Mean ± SD (95% CI)	30.52 ± 0.344 (29.85–31.20)	27.16 ± 0.501 (26.18–28.15)	
Marital status	- Single	140 (37.3)	66 (30.0)	χ^2^ = 3.294 (0.075)
	- Married	235 (62.6)	174 (70.0)	
Education	- Primary	12 (3.2)	0 (0.0)	FET = 13.781 (0.003 *)
	- Secondary	71 (18.9)	50 (22.7)	
	- Bachelor	267 (71.2)	143 (65.0)	
	- Postgraduates	25 (6.7)	27 (12.3)	
Working status	- Student	67 (17.9)	129 (58.6)	χ^2^ = 169.847 (<0.001 **)
	- Working	289 (77.1)	49 (22.3)	
	- Not working	19 (5.1)	42 (19.1)	
Nationality	- Saudi	350 (93.3)	208 (94.5)	χ^2^ = 0.349 (0.602)
	- Non-Saudi	25 (6.7)	12 (5.5)	
Residence	- Central	111 (29.6)	40 (18.2)	χ^2^ = 9.088 (0.011 *)
	- Eastern	88 (23.5)	69 (31.4)	
	- Western	75 (20.0)	61 (27.7)	
	- Northern	64 (17.1)	20 (9.1)	
	- Southern	37 (9.9)	30 (13.6)	
Monthly income adequacy	- Inadequate	130 (34.7)	82 (37.3)	χ^2^ = 20.020 (<0.001 **)
	- Adequate	177 (47.2)	118 (53.6)	
	- Adequate and saving	68 (18.1)	20 (9.1)	
Physical health problems	- No	294 (78.4)	162 (73.6)	FET = 21.852 (0.158)
	- Yes	81 (21.6)	58 (26.4)	
Psychological health problems	- No	169 (45.1)	147 (66.8)	
	- Yes	206 (54.9)	73 (33.2)	

χ^2^: Chi-square test; FET: Fisher Exact Test; * significant at *p* ≤ 0.05; ** Significant at *p* ≤ 0.01.

**Table 2 healthcare-13-02158-t002:** Male and female participants’ smoking behavior-related data.

Smoking Data	Categories	Males n (375)	Females n (220)	Test (*p*-Value)
		No. (%)	No. (%)	
Preferred smoking method	- Cigarettes	265 (70.7)	64 (29.1)	FET = 107.821 (<0.001 **)
	- Hookah	58 (15.5)	85 (38.6)	
	- E-cigarettes	50 (13.3)	57 (25.9)	
	- E-hookah	2 (0.5)	14 (6.4)	
Duration (years)	- <1	3 (0.8)	38 (17.3)	FET = 101.051 (<0.001 **)
	- 1–4	98 (26.1)	90 (40.9)	
	- 5–9	59 (15.7)	41 (18.6)	
	- ≥10	215 (57.3)	51 (23.2)	
Regularity	- Regular	326 (86.9)	119 (54.1)	χ^2^ = 80.844 (<0.001 **)
	- Irregular	49 (13.1)	101 (45.9)	
Perceived dependency	- Independent	60 (16.0)	79 (35.9)	χ^2^ = 71.513 (<0.001 **)
	- Low	36 (9.6)	33 (15.0)	
	- Average	87 (23.2)	30 (13.6)	
	- High	82 (21.9)	64 (29.1)	
	- Extremely high	110 (29.3)	14 (6.4)	
Perceived initiation causes	- Stress	154 (41.1)	87 (39.5)	χ^2^ = 15.036 (<0.005 *)
	- Interest	75 (20.0)	43 (19.5)	
	- Inquisitiveness	91 (24.3)	40 (18.2)	
	- Habituation	23 (6.1)	9 (4.1)	
	- Peer pressure	32 (8.5)	41 (18.6)	
A smoking family member	- Yes	262 (69.9)	184 (83.6)	χ^2^ = 14.005 (<0.000 **)
	- No	113 (30.1)	36 (16.4)	
Number of previous trials to quit in the last year	- None	111 (29.6)	70 (31.8)	χ^2^ = 1.751 (0.626)
	- 1–5	155 (41.3)	82 (37.3)	
	- 6–10	34 (9.1)	17 (7.7)	
	- 11≤	75 (20.0)	51 (23.2)	
Previous use of nicotine replacement therapy	- Yes	70 (18.9)	7 (3.2)	χ^2^ = 30.202 (0.000 *)
	- No	304 (81.1)	213 (96.8)	
An earnest desire to quit	- Yes	298 (79.5)	162 (73.6)	χ^2^ = 7.548 (0.023 *)
	- No	77 (20.5)	58 (26.4)	

χ^2^: Chi-square test; FET: Fisher Exact Test; * significant at *p* ≤ 0.05; ** Significant at *p* ≤ 0.01.

**Table 3 healthcare-13-02158-t003:** Smoking Abstinence Self-Efficacy (ASE) Scale.

Items (Provoking Situations)	$\bar{x}$ ± SD (95%CI)	Confidence to Abstain from Smoking in Provoking Situations										$\bar{x}$ ± SD (95%CI)
		Males					Females					
		Not at All	Slight	Moderate	Very	Extreme	Not at All	Slight	Moderate	Very	Extreme	
		No. (%)	No. (%)	No. (%)	No. (%)	No. (%)	No. (%)	No. (%)	No. (%)	No. (%)	No. (%)	
With friends at a party	3.23 ± 0.060 (3.11–3.35)	38 (10.1)	40 (10.7)	161 (42.9)	69 (14.8)	67 (17.9)	58 (26.4)	22 (10.0)	55 (25.0)	68 (30.9)	17 (7.7)	2.84 ± 0.089 (2.66–3.01)
2. When getting up in the morning	3.00 ± 0.081 (2.84–3.16)	114 (30.4)	22 (5.9)	85 (22.7)	57 (15.2)	97 (25.9)	26 (11.8)	14 (6.4)	19 (8.6)	44 (20.0)	117 (53.2)	3.31 ± 0.104 (2.10–2.51)
3. When feeling very anxious and stressed	3.29 ± 0.062 (3.77–4.02)	60 (16.0)	30 (8.0)	86 (22.9)	93 (24.8)	106 (28.3)	40 (18.2)	8 (3.6)	72 (32.7)	44 (20.0)	56 (25.5)	3.31 ± 0.092 (3.13–3.49)
4. Over coffee while talking and relaxing	3.59 ± 0.070 (3.55–3.82)	42 (11.2)	34 (9.1)	65 (17.3)	93 (24.8)	141 (37.6)	63 (28.6)	15 (6.8)	57 (25.9)	65 (29.5)	20 (9.1)	2.84 ± 0.092 (2.66–3.02)
5. When I feel I need a lift	3.41 ± 0.066 (3.28–3.54)	41 (10.9)	39 (10.4)	115 (30.7)	85 (22.7)	95 (25.3)	29 (13.2)	11 (5.0)	55 (25.0)	47 (21.4)	78 (35.5)	3.32 ± 0.099 (2.52–2.91)
6. When I am very angry about something or someone	3.44 ± 0.070 (3.81–4.08)	81 (21.6)	61 (16.3)	45 (12.0)	79 (21.0)	109 (29.1)	67 (30.5)	19 (8.6)	49 (22.3)	58 (26.3)	27 (12.3)	2.81 ± 0.096 (2.62–3.00)
7. With my spouse or close friend who is smoking	3.17 ± 0.066 (3.04–3.30)	57 (15.2)	46 (12.3)	110 (29.3)	101 (26.9)	61 (16.3)	67 (30.5)	23 (10.5)	39 (17.7)	51 (23.2)	40 (18.2)	2.88 ± 0.102 (2.68–3.02)
8. When I realize I have not smoked for a while	2.48 ± 0.065 (2.35–2.60)	108 (28.8)	96 (25.6)	77 (20.5)	72 (19.2)	22 (5.9)	28 (12.7)	23 (10.5)	42 (19.1)	26 (11.8)	101 (45.9)	3.15 ± 0.099 (2.15–2.55)
9. When things are not going my way, and I am frustrated	3.28 ± 0.061 (3.66–3.90)	51 (13.6)	36 (9.6)	104 (27.7)	88 (23.5)	96 (25.6)	71 (32.3)	31 (14.1)	39 (17.7)	33 (15.0)	46 (20.9)	2.78 ± 0.104 (2.58–2.99)
Negative affect $\bar{x}$ ± SD (95%CI)	3.34 ± 0.058 (3.56–3.99)						2.97 ± 0.087 (2.80–3.14)					
Sig.	t = 62.133 (<0.001 **)											
Social positive $\bar{x}$ ± SD (95%CI)	3.33 ± 0.045 (3.27–3.45)						2.85 ± 0.082 (2.69–3.01)					
Sig.	t = 97.931 (<0.001 **)											
Habitual craving $\bar{x}$ ± SD (95%CI)	2.96 ± 0.056 (2.85–3.07)						3.27 ± 0.086 (2.29–2.63)					
Sig.	t = 53.193 (<0.001 **)											
Overall scores $\bar{x}$ ± SD (95%CI)	3.22 ± 0.045 (3.31–3.49)						3.02 ± 0.075 (2.61–2.91)					
Sig.	t = 40.662 (<0.001 **)											

$\bar{x}$ = weighted mean; SD = standard deviation; CI = confidence interval; t = independent sample *t*-test; ** significant at <0.001; ASE: Abstinence Self-Efficacy.

**Table 4 healthcare-13-02158-t004:** Smoking decisional balance scale.

Items	$\bar{x}$± SD (95%CI)	Perceived Pros and Cons of Smoking										$\bar{x}$± SD (95%CI)
		Males					Females					
		Not at All	Slight	Moderate	Very	Extreme	Not at All	Slight	Moderate	Very	Extreme	
		No. (%)	No. (%)	No. (%)	No. (%)	No. (%)	No. (%)	No. (%)	No. (%)	No. (%)	No. (%)	
Smoking cigarettes relieves tension.	3.50 ± 0.070 (3.36–3.64)	40 (10.7)	57 (15.2)	74 (19.7)	84 (22.4)	120 (32.0)	75 (34.1)	22 (10.0)	74 (33.6)	38 (17.3)	11 (5.0)	2.49 ± 0.085 (2.32–2.66)
2. I am embarrassed to have to smoke.	3.47 ± 0.071 (3.33–3.61)	40 (10.7)	64 (17.1)	71 (18.9)	79 (21.1)	121 (32.3)	81 (36.8)	18 (8.2)	37 (16.8)	13 (5.9)	71 (32.3)	2.89 ± 0.115 (2.66–3.11)
3. Smoking helps me concentrate and do better work.	3.15 ± 0.077 (2.99–3.30)	81 (21.6)	51 (13.6)	73 (19.5)	72 (19.2)	98 (26.1)	114 (51.8)	17 (7.7)	18 (8.2)	62 (28.2)	9 (4.1)	2.25 ± 0.097 (2.06–2.44)
4. My cigarette smoking bothers other people.	3.18 ± 0.068 (3.05–3.31)	123 (32.8)	102 (27.2)	84 (22.4)	19 (5.1)	47 (12.5)	106 (48.2)	60 (27.3)	16 (7.3)	13 (5.9)	25 (11.4)	2.05 ± 0.091 (1.87–2.23)
5. I am relaxed and, therefore, more pleasant when smoking.	2.82 ± 0.065 (2.69–2.95)	48 (12.8)	57 (15.2)	121 (32.3)	78 (20.8)	71 (18.9)	70 (31.8)	21 (9.5)	52 (23.6)	21 (9.5)	56 (25.5)	2.87 ± 0.106 (2.66–3.08)
6. People think I am foolish for ignoring the warnings about cigarette smoking.	3.70 ± 0.072 (3.56–3.85)	29 (7.7)	63 (16.8)	73 (19.5)	35 (9.3)	175 (46.7)	48 (21.8)	33 (15.0)	22 (10.0)	60 (27.3)	57 (25.9)	3.20 ± 0.102 (3.00–3.41)
Total Pros $\bar{x}$± SD (95%CI)	3.27 ± 0.059 (3.16–3.59)						2.53 ± 0.086 (2.37–2.71)					
Sig.	t = 7.300 (<0.001 **)											
Total Cons $\bar{x}$± SD (95%CI)	3.18 ± 0.056 (3.07–3.29)						2.71 ± 0.084 (2.55–2.89)					
Sig.	t = 4.852 (<0.001 **)											
Overall $\bar{x}$± SD (95%CI)	3.23 ± 0.033 (3.16–3.29)						2.63 ± 0.066 (2.50–2.76)					
Sig.	t = 9.026 (<0.001 **)											

$\bar{x}$, weighted mean; SD = standard deviation; CI = confidence interval; t = independent sample *t*-test; ** significant at <0.001; DB: decisional balance.

**Table 5 healthcare-13-02158-t005:** Total percent scores of smoking ASE and DB.

	Males (375)	Females (220)	Sig.
	No. (%)	No. (%)	
Total ASE			
- Low	112 (29.9)	102 (46.4)	χ^2^ = 26.904 (<0.001 **)
- Average	186 (49.6)	71 (32.3)	
- High	77 (20.5)	47 (21.4)	
Negative affect			
- Low	112 (29.9)	118 (53.6)	χ^2^ = 89.011 (<0.001 **)
- Average	242 (64.5)	45 (20.5)	
- High	21 (5.6)	57 (25.9)	
Social or positive mood			
- Low	198 (52.8)	144 (65.5)	χ^2^ = 9.106 (0.011 *)
- Average	60 (16.0)	25 (11.4)	
- High	117 (31.2)	51 (23.2)	
Habitual craving			
- Low	256 (68.3)	96 (43.6)	χ^2^ = 6.790 (0.034 *)
- Average	26 (6.9)	79 (35.9)	
- High	93 (24.8)	45 (20.5)	
Total DB			
- Negative	218 (58.1)	162 (73.6)	χ^2^ = 56.273 (<0.001 *)
- Positive	157 (41.9)	58 (26.4)	
Total pros			
- Low	273 (72.8)	118 (53.6)	χ^2^ = 41.718 (<0.001 *)
- Average	35 (9.3)	10 (4.5)	
- High	67 (17.9)	92 (41.8)	
Total cons			
- Low	108 (28.8)	41 (18.6)	χ^2^ = 13.543 (<0.001 *)
- Average	28 (7.5)	33 (15.0)	
- High	239 (63.7)	146 (66.4)	

χ^2^: Chi-square test; * Significant at *p* ≤ 0.05; ** significant at *p* ≤ 0.001; ASE: Abstinence Self-Efficacy; DB: decisional balance.

**Table 6 healthcare-13-02158-t006:** Mean differences in ASE and DB scales by male and female participants’ personal and smoking behavior-related data.

Parameter		ASE			DB		
		Males	Females	Sig	Males	Females	Sig
		$\bar{x}$ ± SD	$\bar{x}$ ± SD		$\bar{x}$ ± SD	$\bar{x}$ ± SD	
Age (years)	20–30	3.40 ± 0.919	2.77 ± 1.163	F = 6.537 *p* = 0.002 *	3.21 ± 0.649	2.56 ± 1.044	F = 6.707 *p* = 0.001 *
	31–40	3.25 ± 0.875	2.37 ± 1.027		3.31 ± 0.609	2.88 ± 0.639	
	41–50	2.90 ± 0.595	2.67 ± 0.763		3.08 ± 0.735	2.65 ± 0.964	
Marital status	Single	3.15 ± 0.904	3.00 ± 0.987	F = 2.113 *p* = 0.096	3.29 ± 0.688	2.76 ± 0.811	F = 5.691 *p* = 0.117
	Married	3.01 ± 0.897	2.98 ± 1.163		3.19 ± 0.618	2.57 ± 1.044	
Education	Primary	3.17 ± 0.609	0.00	F = 5.671 *p* = 0.001 *	3.22 ± 0.205	0.00	F = 3.988 *p* = 0.008 *
	Secondary	3.19 ± 0.816	2.67 ± 1.325		3.21 ± 0.455	2.46 ± 1.126	
	Bachelor	3.20 ± 0.953	3.03 ± 1.085		3.26 ± 0.693	2.70 ± 0.985	
	Postgraduates	2.72 ± 0.581	2.60 ± 0.795		2.90 ± 0.650	2.52 ± 0.572	
Working status	Student	3.25 ± 0.868	2.87 ± 1.077	F = 5.196 *p* = 0.006 *	3.38 ± 0.651	2.77 ± 0.893	F = 19.203 *p* = 0.000 *
	Working	3.20 ± 0.920	2.78 ± 0.906		3.19 ± 0.655	2.74 ± 0.698	
	Not working	3.31 ± 0.725	2.36 ± 1.365		3.28 ± 0.385	2.04 ± 1.285	
Nationality	Saudi	3.27 ± 0.938	2.71 ± 1.163	F = 1.829 *p* = 0.177	3.25 ± 0.651	2.62 ± 1.006	F = 0.729 *p* = 0.394
	Non-Saudi	3.26 ± 0.361	3.39 ± 0.146		2.93 ± 0.491	2.81 ± 0.292	
Residence	Central	3.33 ± 0.968	2.95 ± 1.034	F = 8.183 *p* = 0.000 *	3.18 ± 0.610	2.67 ± 0.768	F = 1.212 *p* = 0.305 *
	Eastern	2.93 ± 0.815	2.55 ± 0.962		3.26 ± 0.529	2.74 ± 1.035	
	Western	3.19 ± 0.818	3.15 ± 0.990		3.21 ± 0.791	2.78 ± 0.945	
	Northern	3.23 ± 0.963	2.94 ± 1.243		3.24 ± 0.682	2.39 ± 0.972	
	Southern	3.29 ± 0.829	2.03 ± 1.310		3.32 ± 0.633	2.14 ± 1.069	
Monthly income adequacy	Inadequate	3.32 ± 0.828	3.02 ± 1.161	F = 6.729 *p* = 0.001 *	3.21 ± 0.626	2.81 ± 0.922	F = 2.846 *p* = 0.059
	Adequate	3.15 ± 0.919	2.60 ± 1.029		3.21 ± 0.698	2.52 ± 0.961	
	Adequate and saving	3.50 ± 0.998	2.59 ± 1.261		3.33 ± 0.529	2.53 ± 1.257	
Physical health problems	No	3.27 ± 0.916	2.84 ± 1.112	F = 2.735 *p* = 0.099	3.26 ± 0.672	2.65 ± 0.873	F = 3.571 *p* = 0.059
	Yes	3.19 ± 0.896	2.52 ± 1.101		3.13 ± 0.535	2.55 ± 1.011	
Psychological health problems	No	3.25 ± 0.948	2.62 ± 1.252	F = 10.390 *p* = 0.001 *	3.31 ± 0.694	2.24 ± 1.053	F = 9.371 *p* = 0.002 *
	Yes	3.29 ± 0.880	3.03 ± 0.702		3.16 ± 0.597	3.00 ± 0.688	
Preferred smoking method	Cigarettes	3.37 ± 0.863	2.51 ± 1.169	F = 5.743 *p* = 0.001 *	3.12 ± 0.595	2.41 ± 0.893	F = 9.177 *p* = 0.000 *
	Hookah	2.64 ± 0.875	2.87 ± 0.883		3.61 ± 0.634	3.00 ± 0.544	
	E-cigarettes	3.39 ± 0.923	2.80 ± 1.224		3.42 ± 0.679	2.55 ± 1.214	
	E-hookah	3.33 ± 0.00	3.15 ± 0.871		1.83 ± 0.000	2.50 ± 0.819	
Duration (years)	<1	3.38 ± 0.000	1.63 ± 0.714	F = 26.463 *p* = 0.000 *	3.50 ± 0.000	2.05 ± 0.1027	F = 17.197 *p* = 0.000 *
	1–4	3.25 ± 0.800	3.12 ± 0.891		3.40 ± 0.762	2.87 ± 0.805	
	5–9	3.09 ± 1.014	3.37 ± 1.174		3.16 ± 0.786	2.38 ± 1.067	
	≥10	3.32 ± 0.930	2.45 ± 0.978		3.23 ± 0.646	2.63 ± 0.982	
Regularity	Regular	3.47 ± 0.847	3.43 ± 0.901	F = 91.766 *p* = 0.000 *	3.34 ± 0.559	3.15 ± 0.735	F = 76.799 *p* = 0.000 *
	Irregular	3.07 ± 0.862	3.33 ± 0.654		3.16 ± 0.499	2.10 ± 0.981	
Perceived dependency	Independent	3.43 ± 0.725	3.27 ± 0.676	F = 73.110 *p* = 0.000 *	3.05 ± 0.666	2.40 ± 0.858	F = 24.055 *p* = 0.000 *
	Low	3.17 ± 0.994	2.66 ± 0.967		3.19 ± 0.888	1.56 ± 1.089	
	Average	3.14 ± 0.599	2.07 ± 0.985		3.28 ± 0.618	3.11 ± 0.618	
	High	2.50 ± 0.670	2.23 ± 0.923		3.19 ± 0.518	3.30 ± 0.580	
	Extremely high	2.61 ± 0.787	2.52 ± 0.981		3.33 ± 0.637	2.59 ± 0.552	
Perceived initiation causes	Stress	3.31 ± 0.900	3.27 ± 0.920	F = 12.801 *p* = 0.000 *	3.26 ± 0.593	2.92 ± 0.837	F = 25.456 *p* = 0.000 *
	Interest	3.14 ± 0.912	2.77 ± 0.715		3.34 ± 0.637	2.92 ± 0.402	
	Inquisitiveness	3.14 ± 0.892	2.98 ± 0.974		3.13 ± 0.849	2.82 ± 0.631	
	Habituation	3.41 ± 0.929	3.38 ± 0.926		3.15 ± 0.363	3.20 ± 0.889	
	Peer pressure	3.49 ± 0.900	1.37 ± 0.662		3.16 ± 0.254	1.38 ± 1.033	
A smoking family member	Yes	3.22 ± 0.894	2.70 ± 1.019	F = 9.353 *p* = 0.002 *	3.23 ± 0.663	2.59 ± 0.916	F = 3.826 *p* = 0.051
	No	3.39 ± 0.941	3.01 ± 1.510		3.21 ± 0.607	2.82 ± 1.266	
Number of previous trials to quit in the last year	None	3.40 ± 0.919	2.44 ± 1.182	F = 6.351 *p* = 0.000 *	3.24 ± 0.453	2.37 ± 1.197	F = 1.764 *p* = 0.153
	1–5	3.23 ± 0.856	3.11 ± 1.102		3.26 ± 0.805	2.77 ± 1.026	
	6–10	2.51 ± 0.827	3.09 ± 0.686		3.19 ± 0.482	2.73 ± 0.549	
	11≤	3.08 ± 0.722	2.62 ± 0.774		3.16 ± 0.585	2.71 ± 0.548	
Previous use of nicotine replacement therapy	Yes	3.38 ± 0.747	3.37 ± 1.103	F = 7.900 *p* = 0.005 *	3.02 ± 0.408	2.38 ± 1.465	F = 0.205 *p* = 0.651
	No	3.25 ± 0.944	2.73 ± 1.113		3.28 ± 0.628	2.63 ± 0.966	
An earnest desire to quit	Yes	3.10 ± 0.864	3.12 ± 0.747	F = 13.850 *p* = 0.000 *	3.19 ± 0.647	2.51 ± 1.101	F = 7.875 *p* = 0.005 *
	No	3.03 ± 1.078	2.62 ± 1.123		3.37 ± 0.625	2.24 ± 0.380	

F: ANOVA test; * significant at *p* ≤ 0.001; ASE: Abstinence Self-Efficacy; DB: decisional balance.

**Table 7 healthcare-13-02158-t007:** Distribution of male and female participants’ earnest quitting desire by personal and smoking behavior-related data.

Parameter		Earnest Quitting Desire (Yes)				Sig.
		Males (298)		Females (162)		
		No.	%	No.	%	
Age (years)	20–30	164	55.0	120	74.1	χ^2^ = 16.791 (0.000 *)
	31–40	105	35.2	30	18.5	
	41–50	29	9.7	12	7.4	
Marital status	Single	101	33.9	43	26.5	χ^2^ = 2.636 (0.064)
	Married	197	66.1	119	73.5	
Education	Primary	12	4.0	0	0.0	χ^2^ = 8.953 (0.030 *)
	Secondary	56	18.8	40	24.7	
	Bachelor	205	68.8	112	69.1	
	Postgraduates	25	8.4	10	6.2	
Working status	Student	50	16.8	92	56.8	χ^2^ = 155.508 (0.000 *)
	Working	231	77.5	28	17.3	
	Not working	17	5.7	42	25.9	
Nationality	Saudi	275	92.3	158	97.5	χ^2^ = 5.222 (0.022 *)
	Non-Saudi	23	7.7	4	2.5	
Residence	Central	76	25.5	24	14.8	χ^2^ = 17.559 (0.002 *)
	Eastern	88	29.5	53	32.7	
	Western	63	21.1	48	29.6	
	Northern	46	15.4	13	8.0	
	Southern	25	8.4	24	14.8	
Monthly income adequacy	Inadequate	111	37.2	57	35.2	χ^2^ = 3.790 (0.150)
	Adequate	143	32.0	90	55.6	
	Adequate and saving	44	14.8	15	9.3	
Physical health problems	No	231	77.5	122	75.3	χ^2^ = 0.287 (0.336)
	Yes	67	22.5	40	24.7	
Psychological health problems	No	150	50.3	110	67.9	χ^2^ = 14.171 (0.000 *)
	Yes	148	49.7	52	32.1	
Preferred smoking method	Cigarettes	216	72.5	42	25.9	χ^2^ = 96.073 (0.000 *)
	Hookah	37	12.4	43	26.5	
	E-cigarettes	43	14.4	68	42.0	
	E-hookah	2	0.7	9	5.6	
Duration of smoking (years)	<1	3	1.0	35	21.6	χ^2^ = 66.709 (0.000 *)
	1–4	96	32.2	57	35.2	
	5–9	47	15.8	24	14.8	
	≥10	152	51.0	46	28.4	
Regularity	Regular	215	72.1	60	37.0	χ^2^ = 69.884 (0.000 *)
	Irregular	83	27.9	102	63.0	
Perceived dependency	Independent	55	18.5	69	42.6	χ^2^ = 68.404 (0.000 *)
	Low	23	7.7	31	19.1	
	Average	48	28.2	23	14.2	
	High	45	15.1	27	16.7	
	Extremely high	91	30.5	12	7.4	
Perceived initiation causes	Stress	134	45.0	60	37.0	χ^2^ = 42.248 (0.000 *)
	Peer pressure	17	5.7	41	25.3	
	Interest	55	18.5	34	21.0	
	Inquisitiveness	72	24.2	20	12.3	
	Habituation	20	6.7	7	4.3	
A smoking family member	Yes	204	68.5	131	80.9	χ^2^ = 8.164 (0.003 *)
	No	94	31.5	31	19.1	
Previous use of nicotine replacement therapy	Yes	69	23.2	7	4.3	χ^2^ = 26.990 (0.000 *)
	No	229	76.8	155	95.7	
Perceived self-efficacy	Low	103	34.6	113	69.8	χ^2^ = 16.861 (0.002 *)
	Average	117	39.3	33	20.4	
	High	78	26.2	16	9.9	
Decisional balance	Negative	126	42.3	115	71.0	χ^2^ = 34.671 (0.000 *)
	Positive	172	57.7	47	29.0	

χ^2^: Chi-square test; * significant at *p* ≤ 0.001.

**Table 8 healthcare-13-02158-t008:** Logistic regression analysis of predictors of negative DB, high smoking ASE, and earnest quitting desire.

Predictors	Negative DB		High ASE		Earnest Quitting Desire	*p*-Value	
	AOR (95% CI)	*p*-Value	AOR (95% CI)	*p*-Value	AOR (95% CI)		
Age (in years)							
20–30	0.40 (0.13–1.20)	0.100	0.27 (0.07–1.04)	0.056	0.52 (0.15–1.81)	0.303	
31–40	0.20 (0.07–0.59)	0.004 *	0.87 (0.25–3.01)	0.870	1.28 (0.36–4.56)	0.703	
41–50	Ref		Ref		Ref		
Gender							
Male	0.05 (0.02–0.11)	0.000 *	1.22 (0.53–2.82)	0.642	1.68 (0.75–3.76)	0.209	
Female	Ref		Ref		Ref		
Education							
Primary	1.07 (0.09–0.88)	0.103	1.08 (0.54–2.47)	0.503	0.98 (0.29–1.78)	0.101	
Secondary	Ref		Ref		Ref		
Bachelor	1.54 (0.14–3.11)	0.003 *	2.55 (1.26–4.11)	0.021 *	1.39 (0.66–2.78)	0.091	
Postgraduates	1.01 (0.51–2.16)	0.150	0.86 (0.72–3.44)	0.106	0.79 (0.44–1.91)	0.123	
Income status							
Inadequate	1.13 (0.52–2.46)	0.763	0.70 (0.23–1.14)	0.527	1.72 (0.69–4.26)	0.242	
Adequate	0.78 (0.37–1.66)	0.516	1.60 (0.51–2.08)	0.423	1.85 (0.80–4.26)	0.150	
Adequate and saving	Ref		Ref		Ref		
Residence region							
Central	0.46 (0.09–0.88)	0.024 *	0.37 (0.06–1.41)	0.103	2.17 (0.75–4.22)	0.120	
Eastern	0.21 (0.13–0.81)	0.000 *	0.29 (0.11–0.92)	0.000 *	2.13 (0.56–4.21)	0.171	
Western	Ref		Ref		Ref		
Northern	0.34 (0.25–1.56)	0.165	0.67 (0.45–2.78)	0.641	1.17 (0.86–3.88)	0.104	
Southern	0.12 (0.03–1.95)	0.241	0.51 (0.12–2.19)	0.167	1.02 (0.09–1.61)	0.301	
Physical health problems							
Yes	1.41 (0.97–3.37)	0.226	0.52 (0.20–1.26)	0.140	1.48 (1.06–2.91)	0.023 *	
No	Ref		Ref				
Psychological health problems							
Yes	1.82 (0.86–2.44)	0.168	0.74 (0.11–0.97)	0.014 *	1.68 (0.87–3.24)	0.121	
No	Ref		Ref				
Preferred smoking method							
Cigarettes	1.18 (0.41–2.12)	0.187	0.36 (0.05–2.47)	0.298	1.00 (0.14–1.37)	0.998	
Hookah	0.73 (0.05–1.07)	0.068	0.12 (0.02–0.85)	0.034 *	0.06 (0.01–0.58)	0.030 *	
E-cigarettes	1.33 (0.17–1.13)	0.209	1.17 (0.15–2.31)	0.880	0.58 (0.07–1.87)	0.616	
E-hookah	Ref		Ref		Ref		
Smoking duration							
<1	0.435 (0.123–1.532)	0.195	1.41 (1.06–0.76)	0.008 *	2.95 (0.55–5.79)	0.207	
1–4	0.108 (0.038–0.303)	0.097	0.94 (0.32–2.80)	0.991	1.43 (0.61–3.35)	0.417	
5–9	1.31 (1.07–3.24)	0.006 *	0.42 (0.26–1.96)	0.915	0.88 (0.34–2.28)	0.798	
≥10	Ref		Ref		Ref		
Regularity of smoking							
Regular	2.79 (1.53–3.24)	0.000 *	1.45 (0.22–2.81)	0.469	0.17 (0.05–0.55)	0.003 *	
Irregular	Ref		Ref		0.40 (0.13-0.24)	0.114	
Perceived smoking dependency							
Independent	1.36 (0.31–3.78)	0.696	0.38 (0.07–1.18)	0.404	1.23 (0.30–4.99)	0.774	
Low	1.34 (0.23–1.82)	0.744	0.22 (0.02–3.05)	0.097	0.35 (0.10–1.131)	0.119	
Average	0.27 (0.07–1.29)	0.098	0.15 (0.01–0.54)	0.634	2.81 (0.85–4.27)	0.089	
High	2.67 (1.41–3.69)	0.005 *	0.58 (0.13–0.74)	0.021 *	0.11 (0.04–0.30)	0.000 *	
Extremely high	Ref		Ref		Ref		
A smoking family member							
Yes	0.61 (0.20–1.87)	0.387	0.69 (0.33–1.43)	0.314	0.85 (0.39–1.83)	0.671	
No	Ref		Ref		Ref		
No. of previous quitting trials in the last year							
None	1.10 (0.44–2.71)	0.841	1.97 (0.75–5.18)	0.169	0.30 (0.10–1.91)	0.133	
1–5	0.58 (0.25–1.27)	0.216	0.32 (0.12–1.83)	0.120	1.51 (0.55–4.20)	0.352	
6–10	2.06 (0.63–6.78)	0.232	0.27 (0.08–0.91)	0.035 *	1.23 (1.08–4.07)	0.023 *	
11≤	Ref		Ref		Ref		
Previous use of nicotine replacement therapy							
Yes	2.19 (1.34–3.58)	0.002 *	0.54 (0.26–1.15)	0.108	0.08 (0.02–0.34)		0.000 *
No	Ref		Ref		Ref		
Smoking ASE							
Low	8.33 (3.89–15.17)	0.000 *			0.60 (0.24–0.72)	0.021 *	
Average	4.68 (2.19–10.03)	0.000 *			0.91 (0.35–2.47)	0.855	
High	Ref				Ref		
Smoking DB							
Negative			0.10 (0.04–0.45)	0.000 *	0.21 (0.08–0.58)	0.002 *	
Positive			Ref				
Earnest quitting desire							
Yes	2.75 (1.35–3.61)	0.022 *	2.44 (1.03–5.81)	0.044 *			
No	Ref		Ref				
^2^ Log likelihood (462.347) Cox and Snell R^2^ (0.45) Nagelkerke R^2^ (0.606) (*p* < 0.001)			^2^ Log likelihood (408.572) Cox and Snell R^2^ (0.491) Nagelkerke R^2^ (0.660) (*p* < 0.001)		^2^ Log likelihood (411.308) Cox and Snell R^2^ (0.315) Nagelkerke R^2^ (0.480) (*p* < 0.001)		

Covariates: nationality, marital status, and working status. * Significant at *p* ≤ 0.001. ASE: Abstinence Self-Efficacy; DB: decisional balance; AOR: adjusted odds ratio; CI: confidence interval; Ref.: reference group.

## Data Availability

The current study data are available from the corresponding author upon reasonable request.

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
