# Peer review of "Smoking Abstinence Self-Efficacy, Decisional Balance, and Quitting Desire Among Adult Smokers in Saudi Arabia: Gender-Based Cross-Sectional Study"

_healthcare, 2025, doi:10.3390/healthcare13172158_

Round 1
Reviewer 1 Report
Comments and Suggestions for Authors
Please see the attachment.

Author Response
Thank you very much for your thoughtful and constructive feedback. We appreciate the time and effort you have dedicated to reviewing our manuscript, and your comments have been invaluable in improving the quality of our work. We have carefully considered your suggestions and made the necessary revisions accordingly, as in the attached PDF file.

Reviewer 2 Report
Comments and Suggestions for Authors
This is a well-constructed and thorough article. The following comments are minor recommendations for greater clarity and more resolution in the findings.
------
Line 54: Please include at least one citation to defend the statement that traditional cigarettes are still the most widespread form of tobacco use.
Line 62: Please include a citation for the percentage referenced from the WHO data.
Line 62 (continued): Please define indirect tobacco use - is this solely secondhand, thirdhand, or a combination of the two?
Line 79: Please define the most prevalent etiologies of cancer attributable to tobacco use - these have also varied in past years, evolving (at least in the US) from N/SCLC to adenocarcinoma.
Line 84: Please include a reference to the statement of gender differences across tobacco-mediated COPD.
Line 86: Please expand on the role of gender hormones and tobacco; are these protooncological properties limited to solely estrogen and testosterone, or are other androgens involved as well?
Line 90: Please expand on the type of "co-addictions" seen in other populations. In North America these may be opiates, alcohol, caffeine, or other substances - what specific co-addictive substances are seen internationally?
Line 146: A request for clarity here in reference to the epidemic of smoking. Is the assertion in Line 146 that the epidemic the act of cigarette smoking in of itself (which is in decline, at least in North America), or is the epidemic the continued rise in illnesses associated with longterm use (the peaks of which we are still seeing worldwide)?
Line 181: Please expand on why the selection criteria was limited to those participants who were greater than or equal to 20 years of age. This is of concern because many smokers begin much younger - in the data later presented in Table 2, the largest group for duration of use (at over half of the study population) was greater than 10 years.
Line 245: Were LinkedIn, Twitter, and Instagram the only social media sites advertised on, or were there more? Why were each selected? How were users validated as real people versus algorithms (especially on Twitter, which historically has a lot of automated "users")?
Table 1: Please explain why Physical and Psychological Health Problems variables were both organized in the way reported. There is a simple binary (yes/no) mixed with some specific complaints (i.e. Anemia). Further, some are focused on specific bodily systems (i.e. Bronchial asthma) where others are more systemic (Diabetes mellitus). The concern here in reporting in this style is the fragmentation of groups may limit assessment power, resulting in artificially small groups. Perhaps an easier way of reporting in Table 1 is to restructure both variables to match how they were utilized in the log model in Table 8 (simple binary).
Table 8a: Why were Postgraduates the reference population for the log model? Please defend the choice, as according to Table 1, Postgraduates were the smallest groups besides highest level Primary.
Table 8b: Why was the Southern region selected as the reference population?
Table 8c: Regarding the R-squared values, both are only of average fit despite the significant p-value. It may be worth exploring different reference populations (i.e. selecting reference populations drawn from a larger percentage of the pool, building on 8a and 8b above) to assess changes to model fit.
Line 189: There is a reasonable question related to the role of higher education as a positive predictor of ASE when Post-graduates are used as the reference population. One curiosity may be to repeat the model with a different reference population (i.e. secondary education). If both Bachelor and Postgraduate educational attainment groups are significantly higher, this claim will be bolstered. If it is just one or the other, some mild hedging will be needed for this claim.
Line 226: Please expand on the statement "chronic illness like smoking." Related to a point from Line 146, the smoking in of itself is not the issue, the illnesses induced by direct, secondary, or tertiary exposure to tobacco smoke is.
Author Response

(The authors gave the same response as above.)

Reviewer 3 Report
Comments and Suggestions for Authors
This is an original manuscript analyzing individual factors associated with Smoking Abstinence Self-Efficacy, Decisional Balance, and Quitting Desire for tobacco use (cigarette, hooka, shisha) cessation from a gender perspective in the Saudi Arabian population. Studies focused on tobacco cessation, particularly those that employ a comprehensive approach to addressing identified barriers (gender perspective), are crucial for implementing effective public health policies. While significant progress has been made globally in controlling the tobacco epidemic since the ratification of the WHO Framework Convention on Tobacco Control (WHO-FCTC), it is essential for Saudi Arabia and the WHO-FCTC Parties (countries) to provide tangible support to tobacco users (smokers, hookah users) seeking to quit, based on independent scientific evidence.
General Comments:
The manuscript is well-structured and fulfills the requirements of the Reporting of Observational Studies in Epidemiology (STROBE) guidelines. Specific comments will help authors improve the quality of their manuscript for peer review publication.
Specific Comments:
Scale´s validity and reliability: The authors have validated the questionnaire for the study population, which consists of adults in Saudi Arabia. However, it would be methodologically important to provide specific validation data for the following scales: smoking abstinence self-efficacy (ASE), decisional balance (DB), and quitting desire scales for this population.
Results with subheadings aligned with methods and discussion sections: The results section includes valuable and essential information. Since there is a significant amount of data + 7-8 tables, from a formatting perspective, using subheadings (e.g., "Socioeconomic Characteristics of the Study Population") is advisable. This approach creates a precise flow and provides breaks, enabling readers to follow the information presented in both the text and the tables more effectively. The methodology and discussion utilize subheadings, facilitating better organization and making it easier for the reader to follow and grasp the main findings in each analyzed section.
Please check the additional comments in the attachment:

Author Response

(The authors gave the same response as above.)

Round 2
Reviewer 1 Report
Comments and Suggestions for Authors
I would like to thank the authors for the changes and improvements made to the manuscript.
The abstract is well structured, the methodology is now clear, the results are well presented, and the discussion is comprehensive and clearly outlines the main limitation of the study.
I have only two further suggestions for the authors, which do not compromise the fact that the manuscript can be published, but which, if accepted, I believe could further improve the readability of the manuscript:
1. The introduction is still very long. Without removing the changes made by the other reviewer, I believe it could be further summarised. Such a long introduction may discourage the reader's interest somewhat, despite the article being remarkably interesting.
2. Following the changes made, it seems to me that some references do not appear in the correct order. For example, references 70 and 71 appear before reference 68. In addition, the respective references in the bibliography could be reversed. A check on the order and correct match between the number cited in the text and the reference in the bibliography is necessary.
Author Response
Dear Reviewer,
Thanks for your encouraging words; your critical revision is very helpful and improves the quality of our work.
The revised manuscript and our reply are attached.
Regards,

Reviewer 2 Report
Comments and Suggestions for Authors
Thank you for your edits - I have no other substantive edits at this time.
Reviewer 3 Report
Comments and Suggestions for Authors
The updated version of the manuscript incorporates modifications, including new data, information, and references, to address all specific comments from Reviewer 3 (marked in green).
The manuscript now presents a stronger methodological perspective; it is more coherent and effectively balances both strengths and weaknesses. Valid, up-to-date references support the entire manuscript.
Round 3
Reviewer 1 Report
Comments and Suggestions for Authors
All requests were responded to appropriately. I recommend that the manuscript be published.